# Effect of chiropractic care on low back pain for active-duty military members: Mediation through biopsychosocial factors

Zacariah K. Shannon[1]*, Cynthia R. Long[1], Elizabeth A. Chrischilles[2], Christine M. Goertz[3], Robert B. Wallace[2], Carri Casteel[4], Ryan M. Carnahan[2]

1 Palmer Center for Chiropractic Research, Palmer College of Chiropractic, Davenport, IA, United States of America, 2 Department of Epidemiology, College of Public Health, University of Iowa, Iowa City, IA, United States of America, 3 Department of Orthopaedic Surgery, Duke University, Durham, North Carolina, United States of America, 4 Department of Occupational and Environmental Health, College of Public Health, University of Iowa, Iowa City, IA, United States of America

* Zacariah.Shannon@palmer.edu

## Abstract

This study evaluates biopsychosocial factors as mediators of the effect of chiropractic care on low back pain (LBP) intensity and interference for active-duty military members. Data from a multi-site, pragmatic clinical trial comparing six weeks of chiropractic care plus usual medical care to usual medical care alone for 750 US active-duty military members with LBP were analyzed using natural-effect, multiple-mediator modeling. Mediation of the adjusted mean effect difference on 12-week outcomes of PROMIS-29 pain interference and intensity by 6-week mediators of other PROMIS-29 physical, mental, and social health subdomains was evaluated. The effect difference on pain interference occurring through PROMIS-29 biopsychosocial factors (natural indirect effect = -1.59, 95% CI = -2.28 to -0.88) was 56% (95% CI = 35 to 96) of the total effect (-2.82, 95% CI = -3.98 to -1.53). The difference in effect on pain intensity occurring through biopsychosocial factors was smaller (natural indirect effect = -0.32, 95% CI = -0.50 to -0.18), equaling 26% (95% CI = 15 to 42) of the total effect (-1.23, 95% CI = -1.52 to -0.88). When considered individually, all physical, mental, and social health factors appeared to mediate the effect difference on pain interference and pain intensity with mental health factors having smaller effect estimates. In contrast with effects on pain interference, much of the effect of adding chiropractic care to usual medical care for US military members on pain intensity did not appear to occur through the PROMIS-29 biopsychosocial factors. Physical and social factors appear to be important intermediate measures for patients receiving chiropractic care for low back pain in military settings. Further study is needed to determine if the effect of chiropractic care on pain intensity for active-duty military occurs through other unmeasured factors, such as patient beliefs, or if the effect occurs directly.

**Data Availability Statement:** A de-identified limited dataset of requested variables was provided for

this work by Palmer College of Chiropractic under a data use agreement that prohibits data sharing. The authors of this work did not have special data access privileges. Data are available for researchers who meet the criteria for access by contacting Palmer College of Chiropractic at palmer-research@palmer.edu.

**Funding:** The author(s) received no specific funding to conduct this work. This work received the NCMIC Early Career Award at the 2023 WFC Biennial Congress."

**Competing interests:** The authors have declared that no competing interests exist.

## Introduction

Low back pain (LBP) is a global public health burden [1], common amongst the US general population [2], and impactful for people with physically demanding occupations [3]. Similar to other physically demanding occupations, US military members have a high risk of musculo-skeletal pain, making it one of the most common reasons for interruption of combat duty [4–6]. Clinical practice guidelines recommend noninvasive and nonpharmacologic treatments as first-line therapies for LBP [7–9] which are commonly delivered as part of multi-modal chiro-practic care [10]. Adding chiropractic care to usual medical care for US active-duty military members has been demonstrated to result in greater improvement in pain, disability, physical function, sleep disturbance, fatigue, anxiety, depression, and satisfaction with participation in social roles [11, 12]. The mechanisms of improvement in pain and quality-of-life when multi-modal chiropractic care is added to usual medical care are unknown, however.

Changes in pain are thought to be a result of a confluence of mechanisms [13]. To better understand a cause, it can be broken down into a set of component causes that are sufficient to produce the outcome [14]. Considering mechanisms in isolation limits the ability to under-stand how mechanisms fit into a component cause framework. The biopsychosocial model [15] may help elucidate pain by identifying physical, mental, and social factors as components of pain. The biopsychosocial model does not, however, provide a causal framework for evalua-tion. Therefore, it may be necessary to explore individual factor contributions as part of the larger framework when evaluating mechanisms. Mediation analysis is used to evaluate mecha-nisms of a treatment by determining how much of the effect of the treatment on the outcome occurs through intermediary variables [16].

Understanding the mechanisms of multi-modal interventions is important to improve implementation into practice [17]. Mechanisms with large effects inform how the multi-modal intervention is currently achieving effect on outcomes, leading to opportunities to improve intervention delivery by further focusing on components targeting mechanisms with large effects and/or by better addressing potentially important mechanisms showing small effects. Evaluating the mechanisms of multi-modal interventions, such as chiropractic care, is also important if it is to be understood pragmatically, that is, when trying to understand how effects are expected to occur when patients seek care from or are referred to a chiropractor in clinical practice. Mediation analyses have been conducted to evaluate mechanisms of other multi-modal interventions such as tai-chi [18], yoga [19, 20], and physical therapy [20, 21]. The mechanisms of chiropractic care's effect on pain have not been adequately evaluated while incorporating multiple, simultaneous component causes or in the context of the biopsychoso-cial model.

The objective of this study is to evaluate physical, mental, and social health factors as mech-anisms of the difference in effect on LBP of usual medical care plus chiropractic care vs. usual medical care alone for active-duty military members. This study quantifies the effects of an integrative health approach to better understand how change in biopsychosocial factors con-tribute to change in physical pain. Our primary research question was "How much of the effect of chiropractic care on pain interference and intensity occurs through changes in other physi-cal, mental, and social health factors?"

## Methods

This manuscript is reported according to A Guideline for Reporting Mediation Analyses (AGReMA) guidance [22] based on the EQUATOR framework [23] to enhance quality and transparency.

## Study registration

The trial was approved by the Palmer College of Chiropractic IRB, registered on clinical trials.gov (NCT01692275), and written informed consent was obtained from each participant. A subset of the de-identified dataset generated by the parent trial was used for these analyses. The analysis plan for this manuscript was developed prior to analysis, but not published or registered.

## Study design and source of data

De-identified data from a pragmatic, multi-site, clinical trial comparing usual medical care plus chiropractic care to usual medical care alone were used for these secondary, mediation analyses. The original trial protocol [24], primary outcomes [11], and secondary outcomes [12] have been reported elsewhere. The trial was conducted at three military medical facilities: Walter Reed National Military Medical Center in Bethesda, Maryland, Naval Medical Center San Diego in San Diego, California, and Naval Hospital Pensacola in Pensacola, Florida.

## Participants

Participant screening began on September 28th, 2012. Participants were recruited by medical physician referral, study chiropractic clinician referral, and via posted study recruitment flyers. Participants were included if they were an active-duty military member being treated in the military medical center, between the ages of 18 to 50 years old, and had musculoskeletal LBP of any duration. Participants were excluded if they had bone or joint pathology that was a contraindication to spinal manipulative therapy, such as inflammatory arthropathy, spinal instability, cauda equina syndrome, spinal or paraspinal infection, fracture, or tumors, or if they had radiculopathy needing further evaluation or surgery, recent spinal fracture or surgery, or a diagnosis of post-traumatic stress disorder. The primary study included 6 weeks of treatment and post-treatment follow-up at 12 weeks. The final follow-up was completed on February 13th, 2016.

## Sample size

These mediation analyses were not planned at the time of trial commencement and an a priori powered sample size was not possible. The original trial was powered for each site to detect between-group differences of 1.2 points in pain intensity measured by a 0–10 numerical rating scale and 2.4 points in pain-related disability using the Roland-Morris Disability questionnaire [11]. This resulted in the recruitment of 750 participants.

## Effects of interest

The effects of interest for this study are the total effect, indirect effect, direct effect, and proportion mediated. The total effect is the difference in mean pain interference or pain intensity between usual medical care plus chiropractic care and usual medical care alone groups. The indirect effect is the difference in effect between treatment arms on pain interference or pain intensity that occurred through mediators. The direct effect is the difference in effect between treatment arms on pain interference or pain intensity that did not occur through the mediators evaluated. The proportion mediated is the proportion of total effect attributable to indirect effects.

## Measurement

**Intervention/exposure.** We defined the exposure in our models as assignment to treatment group in accordance with the intention-to-treat principle. In the trial [11, 24], interventions delivered by both medical providers and chiropractic study clinicians were not altered by study procedures. Treatment frequency was determined by study clinicians and chiropractic care was limited to a maximum of 12 visits with a study chiropractor during a 6-week treatment period. Of those allocated to usual medical care plus chiropractic care, 350/375 participants had at least 1 visit with a chiropractor and 266/375 had at least 1 visit with a medical provider. Of those allocated to usual medical care alone, 273/375 had at least one visit with a medical provider.

Chiropractic care is typically a multimodal approach containing a combination of education, passive interventions, active interventions, and self-management strategies [25]. The signature treatment associated with chiropractic care is spinal manipulation, which is delivered to most patients receiving chiropractic care [26]. A more specific description of treatments provided by study clinicians in the parent trial have been described elsewhere [26]. Previous work has described potential mechanisms of individual interventions [13] which may be useful to inform clinicians when to employ a given intervention. Evaluation of the effects of a treatment approach, such as chiropractic care, may be more helpful to inform integrative care approaches, addressing when patients should seek chiropractic care and how patients receiving chiropractic care as part of integrative care are likely to respond.

**Patient-Reported Outcomes Measurement Information System (PROMIS) [27, 28].** PROMIS questionnaires were specifically designed to measure patient-reported outcomes encompassing physical, mental, and social health to standardize measures across health domains and to further our understanding of health changes [28]. We selected PROMIS questionnaire responses as appropriate measures for assessing biopsychosocial mechanisms and outcomes of treatment and addressing our research question. PROMIS-29 v1.0 was administered to study participants at baseline, the end of treatment at 6 weeks, and the end of follow up at 12 weeks. The PROMIS-29 questionnaire is a profile of 29 questions including a single-item pain intensity assessment on a 0–10 scale, and 4-question short forms on 5-point Likert scales for each of the following domains: pain interference, physical function, fatigue, sleep disturbance, anxiety, depression, and satisfaction with participation in social roles [29]. Each of the domains are given a T-score normed to a general population average. The general population average is represented by a score of 50 with one standard deviation equal to 10 points [29]. A higher T-score for each domain represents having more of the condition being measured [29]. We transformed physical function and satisfaction with participation in social roles by taking 100-(T-score) to make a higher T-score a consistent indicator of worse health across all PROMIS-29 domains.

**Outcomes.** The outcomes for the mediation analyses are 12-week values of pain interference and pain intensity domains of the PROMIS-29 profile. A description of the outcome measures and select covariates can be found in S1 Table.

**Mediators.** Potential mediators for these analyses are the 6-week values of the remaining PROMIS-29 biopsychosocial domains. Physical health was measured by physical function, fatigue, and sleep disturbance, mental health by anxiety and depression, and social health by satisfaction with participation in social roles.

**Confounding.** The parent trial used adaptive allocation to balance treatment groups on baseline, self-reported sex, age, LBP duration, and LBP intensity. These were the only factors that had a causal relationship with assignment to treatment group and the factors by which models were adjusted to address exposure-mediator and exposure-outcome confounding. The

baseline values of the mediators and outcomes were added as control variables to address potential confounding of the mediator-outcome relationship induced when the baseline values of the mediator or outcome have a causal relationship with the mediator and outcome measured at later follow-up.

## Causal assumptions

Fig 1 shows the assumed causal model for this study. We assumed no mediator-outcome confounders were affected by the exposure.

## Statistical methods

We calculated baseline descriptive statistics and report them by treatment group. We used the CMAverse package [30] in R statistical software (version 4.2.2) [31] to conduct our mediation and sensitivity analyses using the natural effect modeling approach as described by Vansteelandt et al. [32] and the calculation of e-values as described by Haneuse et al. [33].

We assessed exposure-mediator interaction terms for their impact on the model to determine if inclusion was necessary [34]. Exposure-mediator interactions were omitted from the models if they did not change the indirect or direct effect estimates or improve model fit. We used a prespecified plan for handling missing data, assuming the data to be missing at random and imputed values for missing mediator and outcome data by the process of multiple

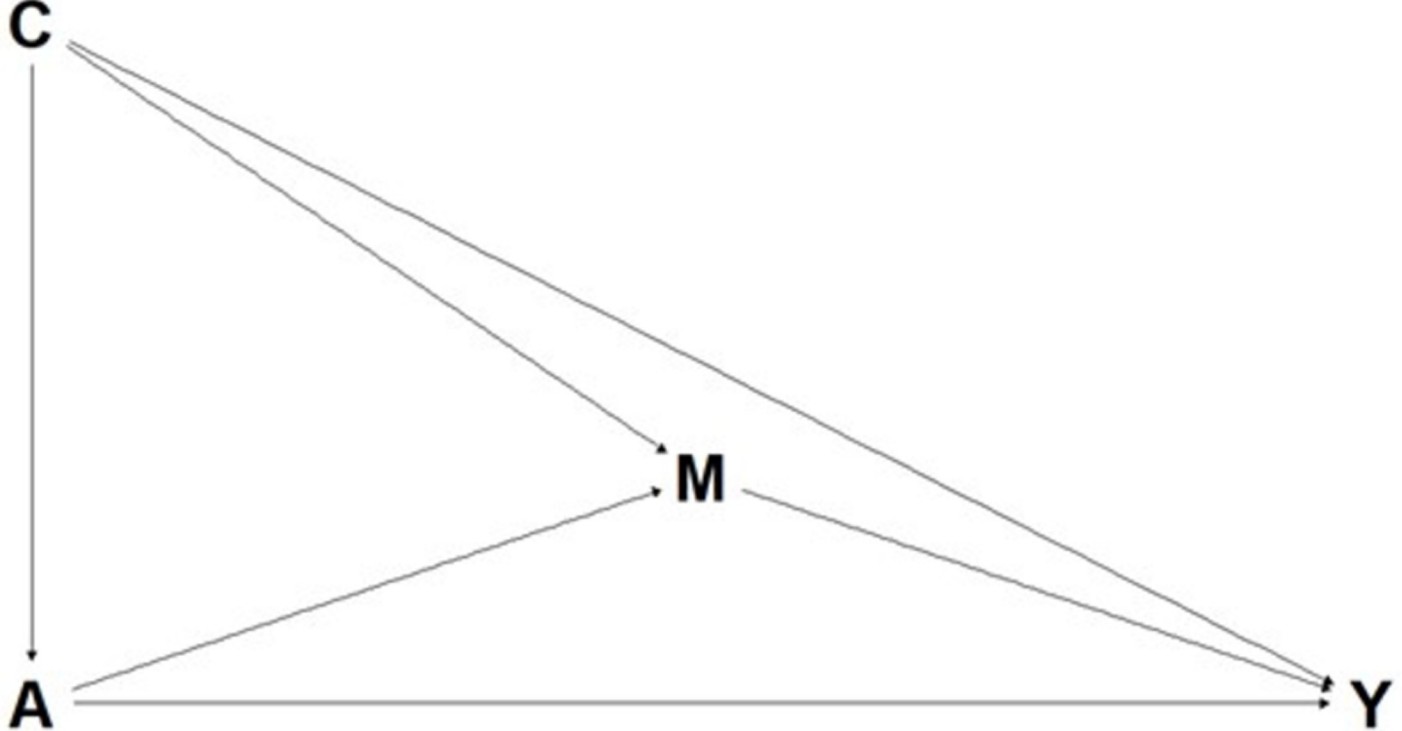

**Fig 1. Directed acyclic graph displaying the causal assumptions of models examining the mediation of the effect of treatment on pain interference (model 1) and pain intensity (model 2).** A (exposure): Usual medical care + chiropractic care vs. Usual medical care alone; M (mediators): Physical function, sleep disturbance, fatigue, anxiety, depression, social roles (6-week values); Y (outcome): Model 1: Pain interference, Model 2: Pain intensity (12-week values); C (confounders): Age, sex, LBP duration, physical function, sleep disturbance, fatigue, anxiety, depression, social roles, pain interference, pain intensity (Baseline values).

imputation by chained equations with 25 imputations and 5 iterations. Standard errors were generated using 1000 bootstrap samples.

We calculated total effect, natural indirect effect, and natural direct effect on the difference scale (usual medical care alone vs. usual medical care plus chiropractic care). We report these effects and the proportion mediated with accompanying standard errors, 95% confidence intervals, and p-values and visualize them in graphical form. The usual care alone group was treated as the control group and the usual medical care plus chiropractic care group as the active group. Therefore, except for proportion mediated, a negative effect estimate shows greater improvement in the usual medical care plus chiropractic care group.

The counterfactual framework of the natural effect model limits the ability to estimate an individual mediator effect in a model containing multiple mediators [35]. When multiple mediators are included in a model, relationships between mediators may impact individual mediator effect estimates. Individual mediator contribution was evaluated in individual models. Then in a sequential addition process [16], we explored if adding mediator variables of smaller individual effect added to the mediation of those showing a larger effect. The Individual-mediator and sequential models were fit for each mediator with the same control variables as the full models.

## Sensitivity analyses

To evaluate the effects of missing data, the models were repeated using complete case analysis. We also calculated e-values to assess the potential impact of unmeasured confounding [33].

## Results

### Participants

Table 1 displays the baseline characteristics of participants enrolled in the trial. Characteristic of the US active-duty military population, the sample was young (mean age = 30.9 years) and predominantly male (76.7%). All participants had a current episode of LBP at baseline. The sample was approximately evenly distributed between baseline duration of three months or less and greater than three months, with more than one-third of the sample having a baseline duration of pain greater than one year. Compared to the general population, the sample mean was better than the population average for both anxiety and depression. Further exploration of anxiety and depression showed that 67% of participants reported 0 for all depression questions and 46% reported 0 for all anxiety questions at baseline. Seventy-nine percent of participants reported less depression symptoms than the general population average and 58% of participants reported less anxiety symptoms. S2 Table shows the completeness of data for mediators and outcomes in total and by treatment arm. Data were more complete for the 6-week measures (84–86%) than the 12-week measures (76–77%).

### Outcomes and estimates

The exposure-mediator interactions had little effect on the estimates of the natural direct and natural indirect effects and did not improve model fit; therefore, they were not included in the final models for either pain interference or pain intensity. Because exposure mediator interactions were not included in the models, the pure natural direct effect and total natural direct effect were equal, and the pure natural indirect effect and total natural indirect effect were equal. For simplicity in reporting, the effects are reported as natural direct effect and natural indirect effect.

**Table 1. Baseline characteristics of participants allocated to usual medical care plus chiropractic care vs. usual medical care alone (n = 750).**

| | UMC (n = 375) | UMC + CC (n = 375) |
|---|---|---|
| Age, years, mean (SD) | 30.8 (8.8) | 30.9 (8.7) |
| Sex, n (%) | | |
| Male | 287 (76.5) | 288 (76.8) |
| Female | 88 (23.5) | 87 (23.2) |
| Race, n (%) | | |
| White | 252 (67.2) | 255(68.0) |
| Black or African American | 72 (19.2) | 77 (20.5) |
| Asian | 20 (5.3) | 10 (2.7) |
| Native Hawaiian or Other Pacific Islander | 2 (0.5) | 7 (1.9) |
| American Indian or Alaska Native | 2 (0.5) | 0 (0) |
| Multi-racial | 8 (2.1) | 6 (1.6) |
| Unspecified | 19 (5.1) | 20 (5.3) |
| Ethnicity, n (%) | | |
| Not Hispanic or Latino | 286 (76.3) | 300 (80.0) |
| Hispanic or Latino | 66 (17.6) | 52 (13.9) |
| Unspecified | 23 (6.1) | 23 (6.1) |
| Current LBP episode duration, n (%) | | |
| < 7 days | 81 (21.6) | 73 (19.5) |
| 7 days to < 16 days | 40 (10.7) | 33 (8.8) |
| 16 days to < 1 month | 23 (6.1) | 37 (9.9) |
| 1 to 3 months | 40 (10.7) | 39 (10.4) |
| > 3 months to < 6 months | 23 (6.1) | 25 (6.7) |
| 6 months to < 1 year | 28 (7.5) | 27 (7.2) |
| 1 year or more | 140 (37.3) | 141 (37.6) |
| Pain intensity, 0–10, mean (SD) | 5.0 (1.9) | 5.0 (1.9) |
| Pain interference, T-score, mean (SD) | 58.9 (7.2) | 60.0 (7.1) |
| Physical function, T-score, mean (SD)* | 56.8 (7.1) | 56.8 (7.3) |
| Sleep disturbance, T-score, mean (SD) | 55.5 (7.6) | 55.0 (7.8) |
| Fatigue, T-score, mean (SD) | 51.5 (9.8) | 51.8 (10.2) |
| Anxiety, T-score, mean (SD) | 48.1 (9.0) | 48.7 (8.7) |
| Depression, T-score, mean (SD) | 45.0 (6.9) | 45.8 (7.1) |
| Social Role, T-score, mean (SD)* | 54.6 (9.1) | 55.1 (8.9) |

*Transformed: 100-(T-score) to make a higher score indicative of worse health; UMC: usual medical care; UMC+CC: usual medical care plus chiropractic care; LBP: low back pain

Table 2 shows the results of the individual-mediator models. In individual mediator models for pain interference, natural indirect effect point estimates were largest for physical function (mean difference = 1.07, 95% CI = -1.84 to -0.55) and social roles (mean difference = -1.06, 95% CI = -1.59 to -0.52) followed by fatigue (mean difference = -0.91, 95% C = -1.29 to -0.36). Ordering of variables by natural indirect effect magnitude was observed to be the same for pain intensity. For each outcome, variables were added by physical, mental, and social health groupings based on largest individual effect estimates. Therefore, our sequential models began with physical health variables, adding social health, and then mental health variables. The proportion mediated of pain interference by physical health variables was 0.53 (95% CI = 0.30 to 0.88). Adding social health increased the estimated proportion mediated to 0.57 (95%

**Table 2. Proportion mediated and total, direct, and indirect effects of chiropractic care on pain interference and pain intensity from individual mediator models.**

| Model Outcome | Mediator | Natural Indirect effect (95% CI) | Natural Direct effect (95% CI) | Proportion Mediated (95% CI) |
|---|---|---|---|---|
| Pain Interference | Physical Function | -1.07 (-1.84 to -0.55) | -1.72 (-2.87 to -0.50) | 0.38 (0.23 to 0.72) |
| | Fatigue | -0.91 (-1.29 to -0.36) | -1.72 (-3.07 to -0.53) | 0.35 (0.15 to 0.65) |
| | Sleep Disturbance | -0.49 (-0.99 to -0.16) | -1.99 (-3.32 to -0.79) | 0.20 (0.07 to 0.48) |
| | Social Roles | -1.06 (-1.59 to -0.52) | -1.57 (-2.92 to -0.43) | 0.40 (0.21 to 0.75) |
| | Anxiety | -0.56 (-1.04 to -0.19) | -2.15 (-3.44 to -0.99) | 0.21 (0.08 to 0.42) |
| | Depression | -0.42 (-0.79 to -0.10) | -2.52 (-3.68 to -1.16) | 0.14 (0.04 to 0.31) |
| Pain Intensity | Physical Function | -0.29 (-0.39 to -0.12) | -0.93 (-1.25 to -0.63) | 0.23 (0.11 to 0.34) |
| | Fatigue | -0.16 (-0.29 to -0.08) | -1.06 (-1.31 to -0.65) | 0.13 (0.07 to 0.27) |
| | Sleep Disturbance | -0.12 (-0.20 to -0.03) | -1.05 (-1.39 to -0.74) | 0.10 (0.03 to 0.18) |
| | Social Roles | -0.21 (-0.32 to -0.10) | -0.99 (-1.27 to -0.62) | 0.17 (0.09 to 0.29) |
| | Anxiety | -0.13 (-0.24 to -0.03) | -1.06 (-1.40 to -0.75) | 0.11 (0.03 to 0.21) |
| | Depression | -0.09 (-0.18 to -0.02) | -1.12 (-1.42 to -0.77) | 0.07 (0.02 to 0.15) |

A negative effect estimate favors the usual medical care plus chiropractic care treatment arm over the usual medical care alone arm.

CI = 0.35 to 0.98), adding mental health variables to physical and social health variables did not increase the proportion mediated. For pain intensity, there was no increase in proportion mediated when social health and mental health variables were added to the physical health variables (proportion mediated = 0.27, 95% CI = 0.14 to 0.40).

Table 3 shows the results of the multiple-mediator models with the pain interference outcome model result graphically displayed in Fig 2 and pain intensity outcome model in Fig 3. The total effect difference on pain interference favored the usual care plus chiropractic care arm (mean difference = -2.82, 95% CI -3.98 to -1.53, p<0.001). The physical, mental, and social health factors included in the models mediated 56.3% (95% CI, 34.5 to 95.7%) of the difference in effect on pain interference. Comparing the plots of the effects in Fig 2 shows that the natural indirect effect on pain interference acting through the mediators was greater than that of the remaining natural direct effect. The total effect on pain intensity also favored the group that received chiropractic care (mean difference = -1.23, 95% CI 1.52 to -0.88, p<0.001). For pain intensity, 26.4% (95% CI, 15.4–41.8%) of the difference in effect was mediated by the six biopsychosocial mediators. In contrast with pain interference for which the natural indirect

**Table 3. Proportion mediated and total, direct, and indirect effects of chiropractic care on pain interference and pain intensity from multiple mediator models.**

| Model Outcome | Effect | Estimate | SE | 95% CI | p-value |
|---|---|---|---|---|---|
| **Pain Interference** | Total Effect | -2.82 | 0.645 | -3.98 to -1.53 | <0.001 |
| | Natural Indirect effect | -1.59 | 0.360 | -2.28 to -0.88 | <0.001 |
| | Natural Direct effect | -1.24 | 0.571 | -2.27 to -0.09 | 0.034 |
| | Proportion mediated | 0.563 | 0.162 | 0.345 to 0.957 | <0.001 |
| **Pain Intensity** | Total Effect | -1.23 | 0.169 | -1.52 to -0.88 | <0.001 |
| | Natural Indirect effect | -0.32 | 0.081 | -0.50 to -0.18 | <0.001 |
| | Natural Direct effect | -0.91 | 0.158 | -1.16 to -0.56 | <0.001 |
| | Proportion mediated | 0.264 | 0.067 | 0.154 to 0.418 | <0.001 |

A negative effect estimate favors the usual medical care plus chiropractic care treatment arm over the usual medical care alone arm. Mediators included in the models were the 6-week values of physical function, fatigue, sleep disturbance, anxiety, depression, and satisfaction with participation in social roles.

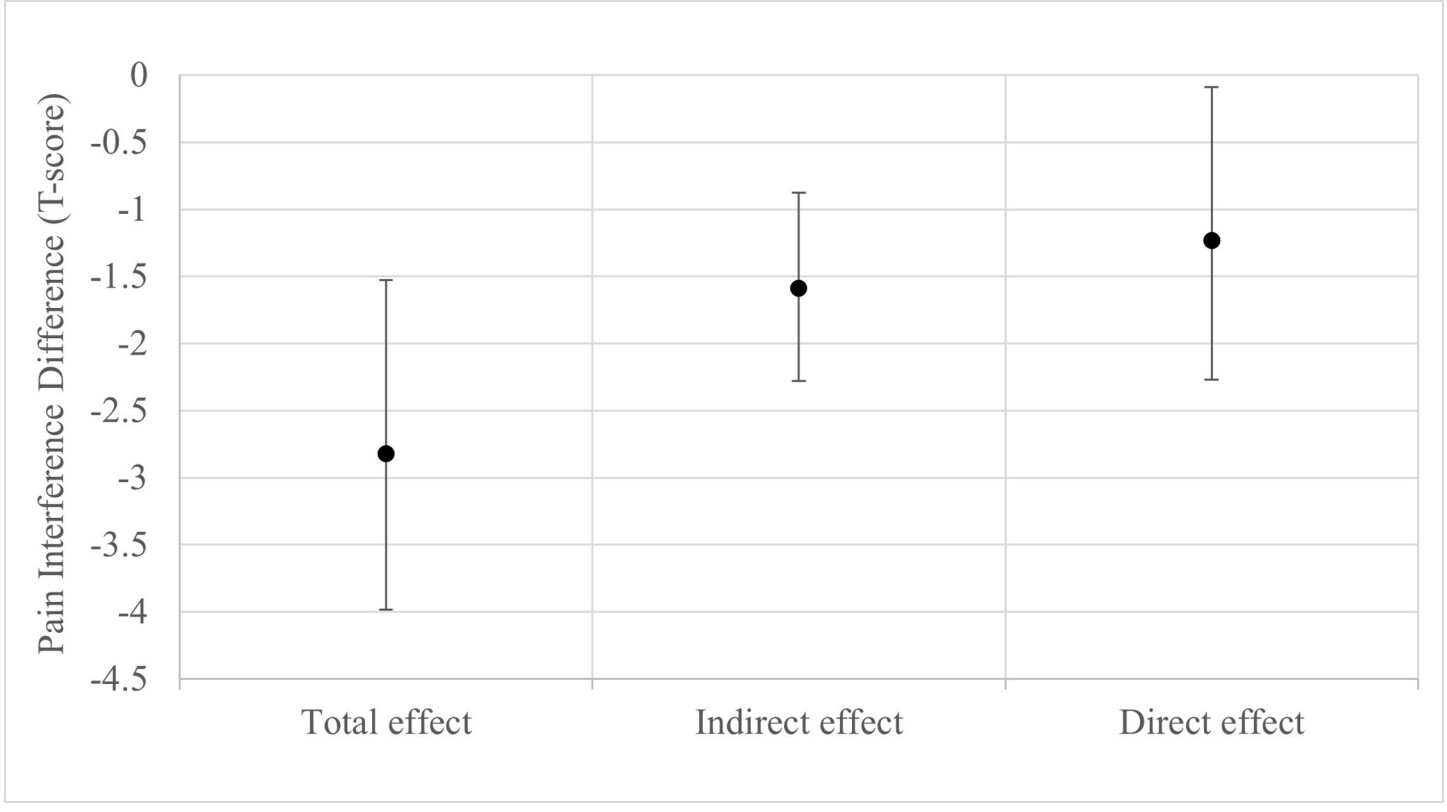

**Fig 2. Display of point estimates and 95% confidence intervals from natural effect multiple-mediator model for the difference in pain interference.** A negative effect estimate favors the usual medical care plus chiropractic care treatment arm.

effect was larger than the remaining natural direct effect, the natural indirect effect on pain intensity was smaller than the natural direct effect (Fig 3).

## Sensitivity parameters

Five hundred and fifty-two participants provided complete data for all mediator and outcome variables while data from 92 participants were missing at both time points. A complete case analysis showed little difference in model parameters compared to our primary analysis which used imputed data for missing mediator and outcome values. The results of the sensitivity analyses for unmeasured confounding can be found in Table 4. For the model with pain interference as the outcome, an unmeasured confounder would need to have an association 2.13 times the total effect observed to nullify the natural indirect effect (E-value RR = 1.62, 95% lower confidence limit = 1.41). The model with pain intensity as the outcome would need an unmeasured confounder 2.45 times the total effect to nullify the natural indirect effect (E-value RR = 1.52, 95% lower confidence limit = 1.33).

## Discussion

In this study of US military members, the addition of chiropractic care to usual medical care had a clinically important [36] and moderate [37] effect on the extent to which pain interfered with daily life and on the intensity of pain. A little over half of the effect of chiropractic care on pain interference appeared to occur through effects on biopsychosocial factors. The largest natural indirect effect estimates in individual models for pain interference were for effects

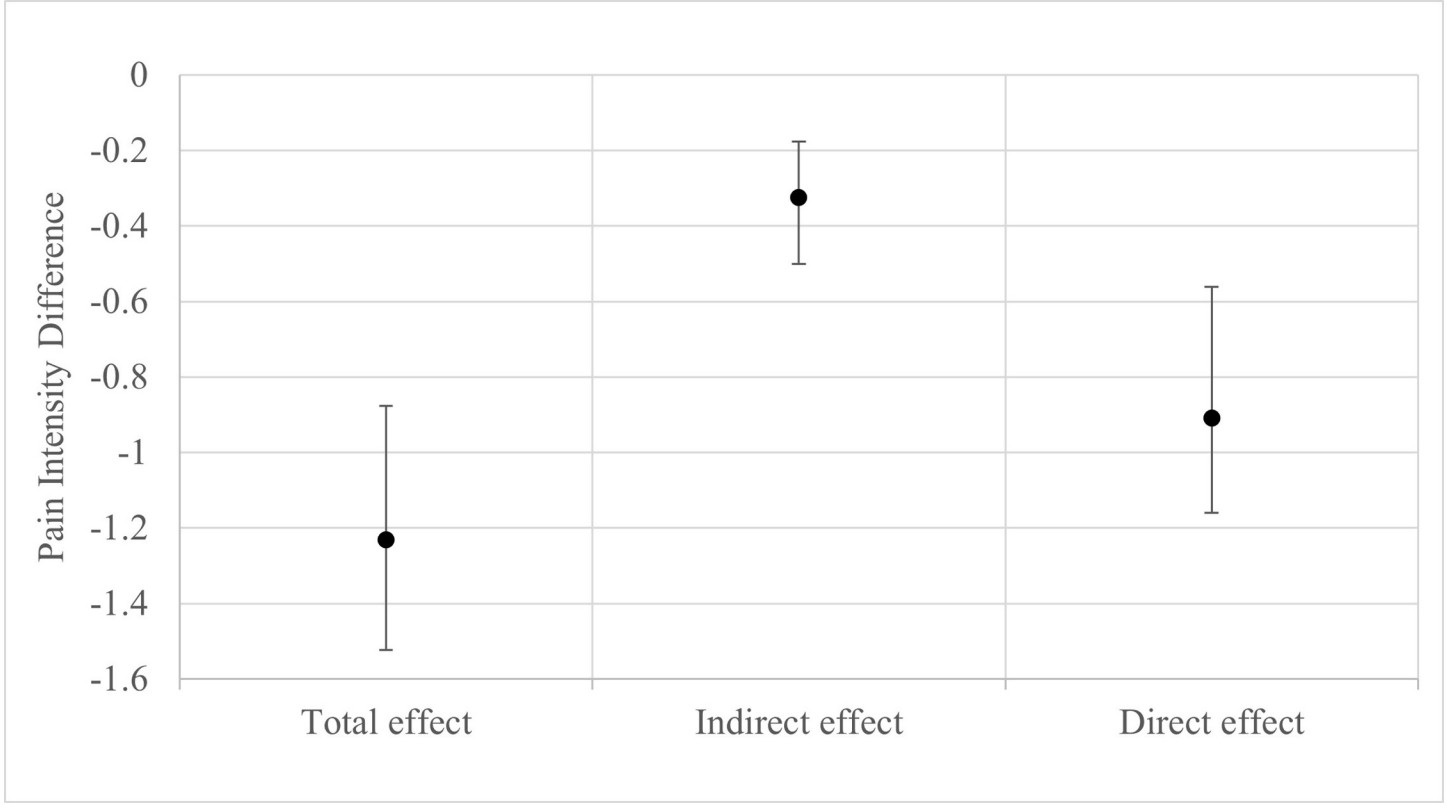

**Fig 3. Display of point estimates and 95% confidence intervals from natural effect multiple-mediator model for the difference in pain intensity.** A negative effect estimate favors the usual medical care plus chiropractic care treatment arm.

occurring through physical function, social activity, and fatigue. The estimate of the natural indirect effect through biopsychosocial factors on pain intensity was smaller than pain interference. The individual models for pain intensity also showed smaller point estimates, with natural indirect effects occurring through physical function being the largest. The natural direct effect estimates for multiple-mediator models of both pain interference and pain intensity outcomes remained statistically significant indicating that the biopsychosocial factors evaluated did not fully mediate the effect of chiropractic care on these outcomes. The natural indirect effect was larger than the natural direct effect in the pain interference model, but not for the pain intensity model.

**Table 4. Risk ratio and e-values for natural direct, natural indirect, and total effects of multiple-mediator models.**

| Model outcome | | Risk ratio | 95% CI | E-value | 95% LCL |
|---|---|---|---|---|---|
| Pain Interference | Total Effect | 0.76 | 0.67 to 0.88 | 1.97 | 1.61 |
| | Natural Indirect Effect | 0.86 | 0.80 to 0.92 | 1.62 | 1.41 |
| | Natural Direct Effect | 0.89 | 0.79 to 0.99 | 1.512 | 1.12 |
| Pain Intensity | Total Effect | 0.62 | 0.55 to 0.71 | 2.601 | 2.19 |
| | Natural Indirect Effect | 0.88 | 0.83 to 0.94 | 1.52 | 1.33 |
| | Natural Direct Effect | 0.70 | 0.63 to 0.79 | 2.193 | 1.84 |

LCL: E-value lower confidence limit

Mediation analyses of trials of other interventions or multi-modal treatment approaches for low back pain have evaluated reassurance [38], treatment expectation [39], pain catastrophizing [18, 21, 40, 41], self-efficacy [41–43], fear avoidance beliefs [42, 44], conscious awareness of body [43], psychological distress [43], perceived stress [43], sleep disturbance [43], and exercise time [43]. Much of this work has focused on patient beliefs in non-military populations. There has been negative or conflicting evidence for various beliefs including fear avoidance beliefs [42, 44] and feeling reassured [38], though differing treatments and methods of evaluation were used. Few mediation studies of interventions for low back pain evaluate simultaneous mechanisms or describe their approach as informed by component cause, model-based theory to inform mediator selection. Sherman et al. demonstrated that self-efficacy, sleep, and hours of back exercise mediated 56.4% of the effect of yoga on disability and that fear, self-efficacy, conscious awareness of body, psychological distress, positive state of mind, and hours of back exercise mediated 49.6% of the effect of stretching on disability [43]. Further study is needed to determine if treatment mechanisms are consistent across interventions or approaches to pain treatment as even similar interventions such as yoga and stretching had variation in magnitude of mediation effects occurring through overlapping, but differing factors [43].

Chiropractic care is a multimodal intervention that commonly includes diagnosis, patient education, passive therapies such as spinal manipulation and other manual therapies, active therapies such as therapeutic exercise, and lifestyle recommendations and self-management strategies [25]. A majority (72%) of participants receiving chiropractic care in this study received a combination of passive and active interventions [26]. The effects of chiropractic care components on biopsychosocial health are still being pieced together. Historically, chiropractic care has encompassed a broad range of components aimed at addressing physical, emotional, and spiritual health [45]. Our current understanding of chiropractic care mechanisms is limited by both the lack of differentiation of component treatments and a lack of understanding the complex contribution of many simultaneous physiological factors [13]. Given the physical nature of many chiropractic care components and a substantial use of therapeutic exercise in this trial, which is often targeted to achieve specific functional improvement [25], effects occurring through physical function may be the least surprising. The effects on pain occurring through effects on fatigue may be due to effects of components such as manual therapies on muscle activity or autonomic activity [13]. Effects occurring through improvement in social health may be related to effects on patient beliefs such fear avoidance beliefs, catastrophizing, or kinesiophobia [13], leading to less fear or greater confidence in the ability to participate in social activity. Future work should differentiate chiropractic care components to better understand these pathways.

The questions asked as part of the PROMIS-29 pain interference domain have conceptual overlap with other domains. The pain interference domain asks to what degree pain interferes with housework, chores, and social activities while physical function asks about the ability to do similar work and satisfaction with social roles asks about the degree to which the participant is satisfied with their ability to do similar work. Changes in one domain may lead to changes in another and mediate this relationship to a greater extent than other outcome domains with less conceptual overlap, such as pain intensity. Correlation induced by conceptual overlap between mediators and the use of multiple mediator models may have decreased the precision of the estimates, producing wide confidence intervals. Changes in quality-of-life measures may also be more likely to mediate changes in other, more complex quality-of-life constructs such as pain interference than more simple measures such as pain intensity. It may be pertinent to evaluate pain intensity as a mediator of these more complex quality-of-life measures than as the outcome in future research.

Using clinical trial data still imposes risk of unmeasured confounding for the mediator-outcome relationship. Our adjustment for physical, mental, and social health factors likely reduced this concern. Further, our sensitivity analysis indicated that it is unlikely that unmeasured confounding would nullify the natural indirect effects observed, giving us confidence that physical and social factors are modest mediators of the effect of chiropractic care on pain interference.

There is limited evidence to inform the tracking of response to treatment for patients receiving chiropractic care, especially for active-duty military members. Current, best practices for chiropractic management of low back pain in adults advise measuring pain intensity, function, and quality-of-life [46]. The findings of this study suggest that PROMIS-29 biopsychosocial factors may be more important intermediates for improvement in pain interference than pain intensity and that physical function and social activity are likely important individual factors. Confirmation of these findings may be useful for chiropractic clinicians and patients in military settings to better understand response to chiropractic care and for the evaluation of treatment plan and prognosis.

The military population under study is distinct from the general population in several ways. While it is generally recognized that LBP often affects the ability to work [1, 3], military personnel face additional risk of losing their ability to work and discharge from service due to loss in function independently associated with LBP and PTSD [4]. This loss in ability is felt at both the individual level and at the level of broader military readiness. The training that military members receive towards being physically and mentally resilient may directly contribute to the current need to target emotional numbness and psychological avoidance to address the large burden of chronic pain and PTSD [47]. Integrative approaches are recommended to address these components [47]. Combining chiropractic care with usual medical care appears to affect some components of broader biopsychosocial measures, but the effect on more nuanced measures of emotional numbness and avoidance remains unknown.

## Limitations

The generalizability of our findings to the general population may be limited due to the study being conducted with US active-duty military members. Compared to the general population, the participants in this sample were predominantly male and reported lower levels of anxiety and depression with most of the sample reporting no to very little mental health symptoms. There are several factors which may have contributed to a low level of reporting mental health symptoms including studying a younger and healthier population than the general population or the associated stigma around mental health symptom reporting and treatment in the US military [48–50]. Fatigue, a recognized indicator of potential mental health problems in general practice [51], was important in both pain interference and pain intensity models. Similar analyses should be conducted in non-military populations, and people reporting mental health symptoms to better understand if mechanisms are consistent in these groups or if they vary by patient characteristics or condition.

Patient beliefs, which were not measured in this study, may be important to understand the mechanisms of chiropractic care. Changes in beliefs, including fear avoidance [40, 44, 52], catastrophizing [18, 21, 41], and self-efficacy [19], have been evaluated as mediators of other complementary health approaches. Similarly, patient beliefs have also been identified as important factors in prognostic risk-based care stratification for LBP care [53–55]. Future research evaluating these components as mechanisms is needed.

The mediators in our analyses were measured at 6 weeks which corresponded to the end of the chiropractic care treatment period, while the outcomes were measured at 12 weeks.

Because of this, we cannot be certain that changes in the mediators explicitly occurred before changes in the outcomes, rather than simultaneously or after improvements in pain. An alternative way to characterize the findings of this study is that our analysis evaluates mediation of sustained improvement at 12 weeks following 6 weeks of chiropractic care.

## Conclusion

US military members likely experience some improvement in low back pain interference and intensity when chiropractic care is added to usual medical care due to biopsychosocial mechanisms. Improvements in physical function and social activity appeared to be the largest contributors. Significant natural direct effects remained in both interference and intensity models, indicating that additional factors, such as patient beliefs, may be important mechanisms needing further evaluation.

## Supporting information

**S1 Table. Description of outcome and select covariate measurements.**
(PDF)

**S2 Table. Completeness of outcome and mediator data.** UMC: usual medical care; UMC +CC: usual medical care plus chiropractic care.
(PDF)

## Author Contributions

**Conceptualization:** Zacariah K. Shannon, Ryan M. Carnahan.

**Formal analysis:** Zacariah K. Shannon.

**Methodology:** Zacariah K. Shannon, Cynthia R. Long, Elizabeth A. Chrischilles, Ryan M. Carnahan.

**Supervision:** Cynthia R. Long, Elizabeth A. Chrischilles, Christine M. Goertz, Robert B. Wallace, Carri Casteel, Ryan M. Carnahan.

**Writing – original draft:** Zacariah K. Shannon.

**Writing – review & editing:** Zacariah K. Shannon, Cynthia R. Long, Elizabeth A. Chrischilles, Christine M. Goertz, Robert B. Wallace, Carri Casteel, Ryan M. Carnahan.

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
