## [Decision Letter · Decision Letter 0]

23 May 2024

PONE-D-24-07242Effect of chiropractic care on low back pain: Mediation through biopsychosocial factorsPLOS ONE

Dear Dr. Shannon,

Thank you for submitting your manuscript to PLOS ONE. After careful consideration, we feel that it has merit but does not fully meet PLOS ONE’s publication criteria as it currently stands. Therefore, we invite you to submit a revised version of the manuscript that addresses the points raised during the review process.

We look forward to receiving your revised manuscript.

Kind regards,

Bernard X.W. Liew

Academic Editor

PLOS ONE

Journal Requirements:

2. For studies involving third-party data, we encourage authors to share any data specific to their analyses that they can legally distribute. PLOS recognizes, however, that authors may be using third-party data they do not have the rights to share. When third-party data cannot be publicly shared, authors must provide all information necessary for interested researchers to apply to gain access to the data. (https://journals.plos.org/plosone/s/data-availability#loc-acceptable-data-access-restrictions) 

**Additional Editor Comments:**

Dear authors,

Thank you for this submission. I have decided to proceed with making a decision based on the assessment from one Reviewer, as I was not able to secure another Review after approaching 10 candidates. Overall the assessment of Reviewer one was positive and I would encourage you to address his comments carefully. In addition to that, I would like the authors be more specific as to what kind of direct and indirect effects have been reported - e.g. natural, pure, total. Also, can the choice of the number of imputations and iterations used in the MICE algorithm be rationalised?

Regards,

Bernard

Reviewers' comments:

Reviewer's Responses to Questions

**Comments to the Author**

1. Is the manuscript technically sound, and do the data support the conclusions?

Reviewer #1: Yes

2. Has the statistical analysis been performed appropriately and rigorously? 

Reviewer #1: Yes

3. Have the authors made all data underlying the findings in their manuscript fully available?

Reviewer #1: Yes

4. Is the manuscript presented in an intelligible fashion and written in standard English?

Reviewer #1: Yes

5. Review Comments to the Author

Reviewer #1: This a very good piece of work and is generally well-written and the statistical analyses are appropriate and well-performed. This is a very publishable piece of work, which will be of interest to readers. The study was well planned, well-conducted, and recruiting 750 partiaipnts to trial is no easy feat, so the authors should be congratulated on these points. I have a few comments that I believe will strengthen the manuscript and add greater value:

Given that the study was conducted with active-duty military personnel, I would expect there to be potentially important differences between active-duty military personnel and the general population. I am aware you brielfly mention this in the limitations, but it is crucial that such potential differences are pointed out throughout the discussion. I would even go as far as to recommend that this is reflected in the title of the paper, which would be consistent with the PICO (population, intervention, control, and outcomes) approach.

The ‘profile’ of potentially important psychological factors (e.g. ‘subclinical’ PTSD symptoms that do not constitute a diagnosis, potential effects of training to be mentally and physically resilient, etc.) and social factors (potentially more independent than civilians, likely relationships between injuries on work ability, career prospectts, etc.) is likely to be different in military personnel compared to civilians, so this also needs to be mentioned explicitly in the discussion , with citations to relevant research. I would also recommend mentioning the potential virtues of performing these analyses in military personnell, especially given that many of the social factors will be similar.

On this point, it might be worth running a quick comparison between the three sites since each may influence the results (different social support, mental health care access, etc): Walter Reed National Military Medical Center in Bethesda, Maryland, Naval Medical Center San Diego in San Diego, California, and Naval Hospital Pensacola in Pensacola, Florida.

Please provide some more detail of what constituted a contraindication for spinal manipulative therapy, as currently this is too vague: "Participants were excluded if spinal manipulative therapy was contraindicated"

Whilst intention to treat is useful for gaining estimates of ‘real life’ effects, for an explanatory analysis such as mediation analysis, we do want to know what actually happened (rather than what should have happened. As you state, "Of those allocated to usual medical care plus chiropractic care, 350/375 participants had at least 1 visit with a chiropractor” Therefore, I would like to see per protocol sensitivity analyses, performed using data on who actually received care, and how many sessions they received, etc. One would expect greater (or different) mediation effects of treatment for those receiving more sessions, for instance.

Also, according to the main trial publication, outcome data were collected at 2, 4, 6, and 12 weeks. And the authors state: "PROMIS-29 v1.0 was administered to study participants at baseline, the end of treatment at 6 weeks, and the end of follow up at 12 weeks.” Therefore, why limit the mediation analyses to the 12 weeks data? Surely, one would expect the earlier time point(s) to provide useful mechanistic information. Please explore this further.

Please expand the discussion with some exploration of why chiropractic care could affect physical function, social activity, and fatigue, from a mechanistic point of view. Fatigue is particularly important here.

6. PLOS authors have the option to publish the peer review history of their article (what does this mean?). If published, this will include your full peer review and any attached files.

Reviewer #1: No

---

## [Author Response · Author response to Decision Letter 0]

8 Jul 2024

To the reviewers and managing editor at PLOS One,

We thank the reviewer and editor for their time in reviewing our manuscript and their helpful suggestions to improve our manuscript. We have included responses to the points raised below. Thank you again for your time.

Editor Comments:

• We have clarified that we are reporting the natural effects and have added description regarding our missing data approach.

Reviewers' Comments:

Given that the study was conducted with active-duty military personnel, I would expect there to be potentially important differences between active-duty military personnel and the general population. I am aware you briefly mention this in the limitations, but it is crucial that such potential differences are pointed out throughout the discussion. I would even go as far as to recommend that this is reflected in the title of the paper, which would be consistent with the PICO (population, intervention, control, and outcomes) approach.

• Thank you for this suggestion. We acknowledge this is an important point and have updated the title to reflect this and have also made additional clarifications in the discussion.

The ‘profile’ of potentially important psychological factors (e.g. ‘subclinical’ PTSD symptoms that do not constitute a diagnosis, potential effects of training to be mentally and physically resilient, etc.) and social factors (potentially more independent than civilians, likely relationships between injuries on work ability, career prospects, etc.) is likely to be different in military personnel compared to civilians, so this also needs to be mentioned explicitly in the discussion , with citations to relevant research. I would also recommend mentioning the potential virtues of performing these analyses in military personnel, especially given that many of the social factors will be similar.

• We have added to the discussion in an attempt to place the profile of questions in the context of the military health system.

Please provide some more detail of what constituted a contraindication for spinal manipulative therapy, as currently this is too vague: "Participants were excluded if spinal manipulative therapy was contraindicated"

• We have added text in an attempt to clarify what is meant by this statement.

Whilst intention to treat is useful for gaining estimates of ‘real life’ effects, for an explanatory analysis such as mediation analysis, we do want to know what actually happened (rather than what should have happened. As you state, "Of those allocated to usual medical care plus chiropractic care, 350/375 participants had at least 1 visit with a chiropractor” Therefore, I would like to see per protocol sensitivity analyses, performed using data on who actually received care, and how many sessions they received, etc. One would expect greater (or different) mediation effects of treatment for those receiving more sessions, for instance.

• There are many facets to discuss surrounding this point. We chose to use the intention to treat approach because we view it as the most rigorous approach to addressing our research question. To get a correctly specified mediation model, there is an assumption of no unmeasured confounding of both the exposure-mediator and mediator-outcome relationships. By using the intention to treat approach we are able to take advantage of the randomization of the parent trial to ensure that there will not be confounding of the exposure-mediator relationship. If we instead take a per-protocol approach then it allows for exposure-mediator confounding when there is a shared cause of participants receiving chiropractic care and change in the biopsychosocial mediator(s) leading to biased models. 

• Nevertheless, we agree with the reviewer that this is an interesting topic and an important concept to study. The dataset we obtained through a data use agreement for this study did not have the number of chiropractic care visits for each participant. This manuscript does not address dosing effects (number of visits) of chiropractic care because we did not think that the original trial design and data collected as part of that study would be adequate to model without severe bias induced by exposure-mediator confounding. We have planned analyses of a subsequent trial (https://reporter.nih.gov/project-details/10248442) that will specifically address the mechanisms related to the number of chiropractic care visits while also further exploring if there are differences in mechanistic effects by treatment components of chiropractic care which were not collected in the trial we are currently reporting on.

• As mentioned in point 1, we feel that the approach we have taken is less biased than a per protocol analysis due to the exposure-mediator confounding. We view our approach as the more conservative option that, if biased, would be towards the null hypothesis/more conservative effect estimates.

Also, according to the main trial publication, outcome data were collected at 2, 4, 6, and 12 weeks. And the authors state: "PROMIS-29 v1.0 was administered to study participants at baseline, the end of treatment at 6 weeks, and the end of follow up at 12 weeks.” Therefore, why limit the mediation analyses to the 12 weeks data? Surely, one would expect the earlier time point(s) to provide useful mechanistic information. Please explore this further.

• The primary outcomes of the trial were pain-related disability and pain intensity and were collected at each of the baseline, 2,4,6, and 12 weeks. We did not use these variables in our analyses. We used the PROMIS-29, which was only collected at baseline, 6, and 12 weeks. We chose the PROMIS-29 because it included constructs of health beyond simple pain measures, including domains consistent with the biopsychosocial model we are attempting to evaluate. We adjusted our models for baseline values of the PROMIS-29 domains, used the 6-week values of the “non-pain” domains of the PROMIS-29 as the mediators, and the 12-week values of the “pain” domains (pain intensity and pain interference) as the outcomes. This time-lag ensured that the mediator changed prior to the outcome. This is best displayed in Figure 1/described in Figure 1 legend (bottom of page 9).

• We describe in the last paragraph of the limitations section that this schedule of measurement may not be ideal and could be alternatively described as mechanisms of sustained improvement following an initial course of chiropractic care. The literature is not clear on the ideal timeframe for measurement of changes in biopsychosocial health as outcomes and especially as mediators of interventions that would give the intervention sufficient time to affect these broader health factors.

Please expand the discussion with some exploration of why chiropractic care could affect physical function, social activity, and fatigue, from a mechanistic point of view. Fatigue is particularly important here.

• We have added to the discussion regarding potential physiological and psychological approaches/mechanisms that chiropractic care may use to address these mechanisms.

---

## [Decision Letter · Decision Letter 1]

4 Sep 2024

Effect of chiropractic care on low back pain for active-duty military members: Mediation through biopsychosocial factors

PONE-D-24-07242R1

Dear Dr. Shannon,

We’re pleased to inform you that your manuscript has been judged scientifically suitable for publication and will be formally accepted for publication once it meets all outstanding technical requirements.

Kind regards,

Bernard X.W. Liew

Academic Editor

PLOS ONE

Additional Editor Comments (optional):

Dear authors,

Having read through your responses, together with Reviewer 1, I am happy for this to be accepted. Thank you for providing a thorough response and this nice work.

Regards,

Bernard

Reviewers' comments:

Reviewer's Responses to Questions

**Comments to the Author**

1. If the authors have adequately addressed your comments raised in a previous round of review and you feel that this manuscript is now acceptable for publication, you may indicate that here to bypass the “Comments to the Author” section, enter your conflict of interest statement in the “Confidential to Editor” section, and submit your "Accept" recommendation.

Reviewer #1: All comments have been addressed

2. Is the manuscript technically sound, and do the data support the conclusions?

Reviewer #1: Yes

3. Has the statistical analysis been performed appropriately and rigorously? 

Reviewer #1: Yes

4. Have the authors made all data underlying the findings in their manuscript fully available?

Reviewer #1: No

5. Is the manuscript presented in an intelligible fashion and written in standard English?

Reviewer #1: Yes

6. Review Comments to the Author

Reviewer #1: I am happy with the changes made by the authors and would like to congratulate them on this excellent work.

7. PLOS authors have the option to publish the peer review history of their article (what does this mean?). If published, this will include your full peer review and any attached files.

Reviewer #1: **Yes: **David William Evans

---

## [Editor Report · Acceptance letter]

20 Sep 2024

PONE-D-24-07242R1 

PLOS ONE

Dear Dr. Shannon, 

I'm pleased to inform you that your manuscript has been deemed suitable for publication in PLOS ONE. Congratulations! Your manuscript is now being handed over to our production team.

Kind regards, 

on behalf of

Dr. Bernard X.W. Liew 

Academic Editor

PLOS ONE